# Treatment Strategies for Advanced Endometrial Cancer According to Molecular Classification

**DOI:** 10.3390/ijms252111448

**Published:** 2024-10-24

**Authors:** Valentina Tuninetti, Alberto Farolfi, Chiara Rognone, Daniela Montanari, Ugo De Giorgi, Giorgio Valabrega

**Affiliations:** 1Department of Oncology, University of Turin, Medical Oncology, Ordine Mauriziano Hospital, 10128 Turin, Italy; dr.ssatuninettivalentina@gmail.com (V.T.); chiara.93rognone@gmail.com (C.R.); giorgio.valabrega@unito.it (G.V.); 2Department of Oncology, IRCCS Istituto Romagnolo per lo Studio dei Tumori (IRST) “Dino Amadori”—IRST S.r.l., 47014 Meldola, Italy; daniela.montanari@irst.emr.it (D.M.); ugo.degiorgi@irst.emr.it (U.D.G.)

**Keywords:** advanced endometrial cancer, immunotherapy, personalized medicine, target therapy

## Abstract

The management of advanced endometrial cancer (EC) has changed in the last few years due to the introduction of a new molecular classification and the approval of immunotherapy. For a long time, carboplatin plus paclitaxel was considered the standard treatment for first-line advanced EC, since the approval of the combination of chemotherapy plus immunotherapy. For patients with recurrent EC, with previous platinum-based chemotherapy, single-agent immunotherapy or in combination with tyrosine-kinase inhibitor (TKI) has been approved according to mismatch repair status. Ongoing trials are exploring the possibility of a chemo-free future for mismatch repair-deficient (dMMR) EC and new molecular targets are under investigation. The treatment paradigm for advanced EC has shifted from standard chemotherapy for all to a more personalized approach. The aim of this review is to provide an updated therapeutic landscape for the management of patients with advanced/metastatic EC according to their disease history and molecular biology.

## 1. Introduction

Since the recent incorporation of genomic characterization into daily clinical practice by The Cancer Genome Atlas (TCGA) that classifies endometrial cancers (ECs) into four categories—POLE ultramutated, microsatellite instability hypermutated (MSI-H), copy-number-low, and copy-number-high [1]—and the advent of immunotherapy, the 5-year survival rates of advanced or recurrent endometrial cancer (EC) have been approximately 17% [2]. Endometrial carcinoma (EC) was previously divided into two subtypes. Type I, which is associated with a favorable prognosis, is primarily represented by endometrioid adenocarcinoma. This subtype is often linked to unopposed estrogen stimulation and frequently follows a progression from endometrial hyperplasia. In contrast, Type II is characterized by significantly poorer 5-year survival rates and is predominantly associated with non-endometrioid histologies. This subtype typically arises in an atrophic endometrium and can develop from intraepithelial carcinoma, which serves as a precancerous lesion [3,4]. The ProMisE (Proactive Molecular Risk Classifier for Endometrial Cancer) was developed to identify similar subgroups using a combination of immunohistochemistry and mutational analysis, replacing MSI-H with the mismatch repair-deficient group (dMMR) and copy-number-high with the p53-abnormal group [5]. 

For years, the standard of care for EC has been the carboplatin–paclitaxel (CP) regimen [2], after the GOG209 study comparing CP to paclitaxel–doxorubicin–cisplatin (TAP) demonstrated the non-inferiority [6] in terms of overall survival (OS, median of 37 vs. 41 months, respectively; hazard ratio (HR), 1.002; 90% CI, 0.9 to 1.12) and progression-free survival (PFS, median of 13 vs. 14 months; HR, 1.032; 90% CI, 0.93 to 1.15). However, since CP demonstrated a better safety profile and an improvement in quality of life (QoL) compared to the other regimen, it became the standard of care. Recent advancements in the field have led to the development of new treatment options for advanced EC. One such option is immunotherapy, which has shown promising results in clinical trials. Additionally, targeted therapies that specifically attack the molecular pathways involved in EC are being explored as potential treatment strategies [4,7,8]. These advancements provide hope for patients who previously had limited options after the failure of platinum-based chemotherapy. 

Moreover, with the introduction of a new TGCA classification, EC is now considered not a single entity but at least four distinct diseases: POLE mut, dMMR/MSI-H, p53 wild-type/copy-number-low (p53 wt), and p53-abnormal/copy-number-high (p53abn) [3,9]. This molecular classification not only presents prognostication capabilities but also appears predictive for adjuvant treatments [10]. In fact, the most recent international guidelines (e.g., ESGO guidelines for EC) recommend classifying EC patients based on their molecular characteristics, since it was demonstrated that, for instance, the POLE-mut subgroup does not need any adjuvant treatment (and a discussion is now open on the necessity to treat POLE-mut patients in higher stages, as discussed below), whereas the p53abn subgroup deserves a worse prognosis and should include or at least evaluate the incorporation of adjuvant chemotherapy also in stage IA.

Given this background, we have decided to write this review to provide an updated therapeutic landscape of EC. For this reason, we extensively reviewed the literature to provide the most recent studies and also reported the most updated results presented at the last international congresses of randomized clinical trials in this setting.

## 2. Treatment Strategies for MSI-H/dMMR EC

Mismatch repair-deficient tumors exhibit a distinct genetic profile, containing 10 to 100 times more mutations compared to mismatch repair-proficient (pMMR) tumors [11]. These tumors are especially prone to mutations within repetitive DNA sequences known as microsatellites, leading to elevated levels of microsatellite instability (MSI-H) [12]. MLH1, MSH2, MSH6, and PMS2 are the genes involved in the MMR pathway. In the hereditary form (Lynch syndrome), one allele of them is mutated in the germline and the second event occurs spontaneously, whereas in sporadic cases, mutations in alleles occur or are epigenetically silenced. Tumors with dMMR typically show a high number of infiltrating lymphocytes and express programmed death ligand 1 (PD-L1) on their tumor cell membrane, secondary to the high number of frameshift mutations, resulting in mutant protein neoantigens and high mutational tumor burden (MTB) [13]. This should be the rationale behind the evidence of the activity of immune checkpoint inhibitors (ICIs) for advanced MSI-H/dMMR cancer, regardless of anatomic tumor location [14]. 

EC very frequently shows an MSI-H/dMMR phenotype varying from 17% to 33% of the cases [15]. This gave the rationale for evaluating the activity of ICIs in second-line treatment after platinum-based chemotherapy [16,17,18]. Pembrolizumab, a humanized monoclonal anti-PD-1 antibody, demonstrated an objective response rate (ORR) of 13% in a population of patients with PD-L1-positive disease and treated at least with two previous lines of therapy [19]. Other early evidence of pembrolizumab’s antitumor activity in this setting resulted in an objective response in eight of fifteen (53%) patients with dMMR EC in a prospective analysis from the KEYNOTE-016 study [20]. 

KEYNOTE-158 was a non-randomized, open-label, multicohort, phase II trial. Eligible patients from cohort D (EC, regardless of MSI-H/dMMR status) and cohort K (any MSI-H/dMMR solid tumor, except colorectal) with previously treated, advanced MSI-H/dMMR EC received pembrolizumab 200 mg once every 3 weeks for 35 cycles. The ORR, the primary endpoint of this study, was 57% in an initial analysis of outcomes among the first 49 patients with MSI-H/dMMR EC enrolled. In the 2022 update of this study [17], the ORR was 48% (95% CI, 37% to 60%), with a median duration of response not reached (2.9–49.7+ months), a median PFS of 13.1 (95% CI, 4.3 to 34.4) months, and a median OS not reached (95% CI, 27.2 months to not reached). On the basis of this trial, pembrolizumab is now reimbursed in dMMR EC previously exposed to platinum-based chemotherapy (no more than two previous lines) in the United States of America (USA), Europe (EU). 

In the phase III trial, KEYNOTE-775 patients received either lenvatinib (20 mg, taken orally once daily) plus pembrolizumab (200 mg intravenously every 3 weeks) or chemotherapy of the treating physician’s choice (doxorubicin or paclitaxel) in the standard arm (Table 1). The primary endpoints were PFS and OS. Although the KEYNOTE-775 trial was not designed or powered to compare lenvatinib plus pembrolizumab with chemotherapy in the dMMR population, the median PFS was 10.7 months vs. 3.7 months for the standard arm (HR, 0.36; 95% CI, 0.23 to 0.57; *p* < 0.001). Since a significant increase in complete response rate was observed with the combination of pembrolizumab with lenvatinib, it may be appropriate to consider this type of treatment for symptomatic dMMR patients or for those who might benefit from rapid tumor shrinkage given the opportunity to use local treatments such as radiotherapy or surgery [18].

Dostarlimab is a humanized IgG4-k monoclonal antibody that binds PD-1 with high affinity, inhibiting the binding of PD-L1 and PD-L2. In the GARNET study [21] (a phase I, single-arm study, with dostarlimab as the monotherapy in patients with advanced and recurrent solid tumors), patients with EC were grouped into two cohorts: dMMR/MSI-H patients (cohort A1) and pMMR/MSS patients (cohort A2). Dostarlimab 500 mg was given to patients every 3 weeks for four cycles, followed by dostarlimab 1000 mg every 6 weeks until progressive disease (PD). The primary endpoints were ORR and duration of response (DOR). With a median follow-up of 16.3 months (IQR 9.5–22.1) in cohort A1 and 11.5 months (IQR 11.0–25.1) in cohort A2, ORR was 43.5% (95% CI, 34.0% to 53.4%), with 11 complete responses (CRs), and 14.1% (95% CI, 9.1% to 20.6%), respectively. In neither cohort was the median DOR reached [16] (Table 1). On the basis of these results, FDA and EMA approved dostarlimab as a monotherapy in patients with recurrent or advanced dMMR/MSI-H EC that progressed on or after treatment with a platinum-containing regimen (no more than two previous lines of therapy). 

Given the evidence of activity of ICIs in second or subsequent lines, it was natural to try to move this kind of treatment to the first line. In this context, the MITO group conducted phase II trials to evaluate avelumab concurrent to CP and as a maintenance treatment after the end of chemotherapy as a first-line therapy for patients with advanced and recurrent EC, irrespective of its molecular classification (Table 1). Although the addition of immunotherapy to standard treatment did not show any advantages in terms of PFS in the intention-to-treat (ITT) population, a significant increase in median PFS in the dMMR population was seen (HR = 0.46, 95% CI = 0.22–0.94) [22], warranting further evaluation. 

In the NRG-GY018 trial (Table 2), pembrolizumab was combined with standard chemotherapy in advanced or recurrent EC patients. In this phase III, double-blind, randomized controlled trial, newly diagnosed measurable disease (stage III or IVA) or IVB or recurrent EC patients were randomly assigned to receive pembrolizumab or placebo (in 6 cycles every 3 weeks, followed by up to 14 maintenance cycles every 6 weeks) in addition to CP. Patients were stratified into two cohorts according to whether they had dMMR or pMMR disease. In the dMMR cohort, the median PFS was not reached for the pembrolizumab arm vs. 7.6 months for the chemotherapy-only arm (HR = 0.30, 95% CI = 0.19–0.48, *p* < 0.001) [23]. Recently, it was reported that the efficacy of pembrolizumab was independent from the mechanism of MMR deficiency: the improvement in median PFS was consistent in both patients with epigenetic alteration (MLH1 promoter methylation) both in patients with mutations [24]. 

Similarly, the RUBY1 trial (Table 2), a randomized phase 3 placebo-controlled trial, demonstrated that dostarlimab given concurrently with CP followed by a maintenance up to 3 years increased PFS at 24 months to 36.1% (95% CI, 29.3–42.9) from 18.1% (95% CI, 13.0–23.9) (HR = 0.64; 95% CI, 0.51–0.80; *p* < 0.001). Among patients with dMMR, PFS at 24 months was 61.4% (95% CI, 46.3–73.0) in the dostarlimab group versus 15.7% (95% CI, 7.2–27.0) in the placebo group (HR = 0.28; 95% CI, 0.16–0.50; *p* < 0.001). 

In the overall population, with a median follow-up of 25.4 months, overall survival (OS) was greater in the dostarlimab group compared to the placebo group (HR = 0.64; 95% CI, 0.46 to 0.87; *p* = 0.0021). However, the findings did not meet the predetermined significance threshold for stopping criteria. Among patients with dMMR, OS at 24 months was 83.3% (95% CI, 66.8 to 92.0) in the dostarlimab group and 58.7% (95% CI, 43.4 to 71.2) in the placebo group (HR = 0.30; 95% CI, 0.13 to 0.70) [25]. Efficacy was independent of histology: dostarlimab increased the PFS in endometrioid, serous, and carcinosarcoma tumors [26]. 

The efficacy of immunotherapy in combination with CP as a first-line treatment was confirmed by the AtTEnd trial (Table 2), in which atezolizumab given concurrently and as a maintenance therapy until progression was administered. In this trial, patients were stratified by histotypes, recurrent disease (at least 6 months of disease-free interval) versus new diagnosis, and MMR status. In all-comers patients, atezolizumab increased the median PFS to 10.1 months (95% CI, 9.4–12.3) from 8.9 months (95% CI, 8.1–9.6) of placebo (HR = 0.74, 95% CI = 0.61–0.91, *p* = 0.0219). In the dMMR population, the median PFS with atezolizumab was not reached compared to 6.9 months (95% CI, 6.2–9.0) of the placebo arm (HR = 0.36, 95% CI = 0.23–0.57, *p* = 0.0005). In the co-primary endpoint of OS in all-comers, a trend in improvement for atezolizumab was observed, despite 24% of patients in the placebo arm receiving subsequent immunotherapy [27].

## 3. POLE Ultramutated Advanced EC

The recent classification of EC by TCGA has notably influenced treatment approaches in early-stage disease. However, there is limited information regarding treatment responses in advanced EC based on molecular characteristics, particularly for the POLE ultramutated group. Because POLE-mutated and dMMR/MSI-H EC subtypes share the high mutational load, that was hypothesized to be the reason for a better response to ICIs. However, very few patients were enrolled in the previously described trials, probably reflecting the very good outcome of this subgroup. Out of 166 patients enrolled in the MITO END-3 trial, 102 had a block available for NGS analysis and just 1 patient showed a POLE mutation [28]. At the ESMO 2023 congress, an analysis of outcomes by molecular classification of the RUBY1 trial was presented. Of the 494 patients enrolled and randomized, mutational data were available for 400 (81.0%), and 5 patients (1.3%) were POLE-mutated. No patients of this subgroup had relapsed at the time of analyses, independent of the treatment received [26]. 

## 4. pMMR Endometrial Cancer 

pMMR/MSS EC patients are a heterogeneous subgroup of patients where single-agent ICI showed less activity than in the dMMR/MSI-H population, with an ORR of about 13% for pembrolizumab, 14.1% for dostarlimab, 6% for avelumab, and 3% for durvalumab [19,21]. Similarly, lenvatinib, a multitargeted tyrosine kinase inhibitor, showed limited efficacy used as a monotherapy in second-line treatment for recurrent EC (ORR, 14.3% [95% CI, 8.8 to 21.4]) [29]. Thus, combination strategies were explored, also supporting the need to identify predictive biomarkers.

The combination of ICIs with chemotherapy was analyzed in the first-line setting of pMMR/MSS EC patients. Pembrolizumab with CP showed a median PFS for the combination arm of 13.1 months vs. 8.7 months of the standard arm (HR, 0.54; 95% CI, 0.41–0.71; *p* < 0.001) [24]. Similarly, among patients with pMMR/MSS tumors of the RUBY trial, the PFS at 24 months was 28.4% (95% CI, 21.2–36.0) in the dostarlimab group and 18.8% (95% CI, 12.8–25.7) in the placebo group (HR, 0.76; 95% CI, 0.59–0.98), with an OS at 24 months of 67.7% (95% CI, 59.8–74.4) and 55.1% (95% CI, 46.8–62.5), respectively (HR, 0.73; 95% CI, 0.52–1.02) [25]. On the contrary, the median PFS in the pMMR cohort for CP plus atezolizumab was not statistically different from CP plus placebo (HR = 0.92; 95% CI, 0.73–1.16).

Lenvatinib plus pembrolizumab demonstrated efficacy in patients with advanced EC who had previously received at least one platinum-based chemotherapy line [30]. In the ITT population, the median PFS for the combo was 7.2 vs. 3.8 months (HR, 0.56; 95% CI, 0.47 to 0.66; *p* = 0.001) and the OS was 18.3 vs. 11.4 months (HR, 0.62; 95% CI, 0.51 to 0.75; *p* = 0.001).

In the pMMR population, both PFS and OS were significantly longer with lenvatinib plus pembrolizumab than with chemotherapy (median PFS of 6.6 months; 95% CI, 5.6 to 7.4 vs. 3.8 months; 95% CI, 3.6 to 5.0; HR, 0.60; 95% CI, 0.50 to 0.72; *p* < 0.001; and median OS of 17.4 months; 95% CI, 14.2 to 19.9 vs. 12.0 months; 95% CI, 10.8 to 13.3; HR, 0.68; 95% CI, 0.56 to 0.84; *p* < 0.001). The ORR was higher in the pembrolizumab plus lenvatinib group vs. chemotherapy (30.3% vs. 15.1%). In this population, 5.2% of the patients in the lenvatinib–pembrolizumab group and 2.6% of those in the chemotherapy group had a CR. In the pMMR population, the median duration of response was 9.2 months (range from 1.6 to 23.7) with lenvatinib plus pembrolizumab and 5.7 months (range from 0.0 to 24.2) with chemotherapy [18].

Adverse events of grade 3 or higher occurred in 88.9% of lenvatinib plus pembrolizumab patients and 72.7% of chemotherapy patients [18]. The most common serious adverse events were hypertension (in 4.2% of the patients) with lenvatinib plus pembrolizumab and febrile neutropenia (in 4.1%) with chemotherapy. Other frequent adverse events for the combination of the ICI with the TKI were asthenia, weight loss, reduced appetite, stomatitis, and hypothyroidism. Grade 5 adverse events were recorded in 5.7% of the patients receiving lenvatinib plus pembrolizumab and in 4.9% of those receiving chemotherapy. Very recently, data of the LEAP-001 trial (a phase 3 randomized trial of pembrolizumab lenvatinib in the first line) were presented. At the final analysis, this study did not meet its primary endpoint, not improving OS or PFS for the pembrolizumab + lenvatinib arm, for first-line treatment in patients with advanced or recurrent EC, compared to the standard of care involving platinum-based chemotherapy doublets (CP) [31]. Of note, a high rate of grade 5 toxicities was seen in the experimental arm, confirming the need for a close patient follow-up, in particular during the first two months of treatment, in order to early detect changes in blood pressure and/or other adverse events that may ultimately translate to treatment-related deaths.

## 5. p53 Wild-Type Endometrial Cancer

All the trials with immunotherapy in the first line had a combination phase, followed by a maintenance phase where immunotherapy was given alone. The shape of the curves presented were quite similar, independent of the ICI used with the curves overlapping in the first months when the patients were treated concomitantly with CP and immunotherapy, and diverging after the maintenance phase initiation. This observation has raised the question of whether the efficacy is related to the priming phase of chemotherapy or whether the maintenance phase could be sufficient. In this context, other trials investigated the role of maintenance treatment in EC.

The SIENDO trial was a randomized, double-blind, placebo-controlled trial with selinexor as the oral maintenance treatment given after a chemotherapy response. Selinexor is a targeted therapy classified as a “selective inhibitor of nuclear export” (SINE). SINEs inhibit the exportin 1 (XPO1) protein, which plays a crucial role in transporting proteins out of the cell nucleus. This includes the export of tumor suppressor proteins that are essential for inhibiting cancer cell proliferation. By blocking XPO1, selinexor enhances the activity of these tumor suppressor proteins, thereby contributing to the suppression of cancer cell growth. One notable tumor suppressor protein affected by this mechanism is p53, encoded by the TP53 gene. In the SIENDO trial, PFS for selinexor did not meet the threshold for statistical significance in the ITT population [28]; however, at the ASCO meeting in June 2023, an analysis in TP53 wild-type EC patients was presented. In this analysis, 77 EC p53 wild-type patients received maintenance therapy with selinexor and 36 patients received placebo. At a median follow-up of 25.3 months, selinexor significantly delayed cancer growth for a median of 27.4 months versus 5.2 months with placebo. In order to confirm this finding, a phase III trial with selinexor as a maintenance treatment in patients with p53 wild-type, advanced, or recurrent EC is currently ongoing (XPORT-EC-042 trial—NCT05611931) [32].

## 6. p53-Abnormal Endometrial Cancer

p53-abnormal EC patients presents a bad prognosis in the early stages and deserve treatment intensification according to the new ESGO guidelines. In advanced EC, little data are available on treatment response according to molecular data, particularly when immunotherapy is used. A preliminary result of the GARNET study in the second-line setting, with dostarlimab, showed fewer responses to the drug in patients carrying the TP53 mutation [28]. One of the first studies to combine new target agents as the first-line treatment with chemotherapy in patients with advanced EC was the GOG-86P trial, a three-arm, randomized phase II study of PC plus bevacizumab, or PC plus temsirolimus, or ixabepilone and carboplatin plus bevacizumab [33]. In a retrospective analysis of p53 status in this study, patients with an overexpression of p53 by IHC had a better PFS with bevacizumab compared with temsirolimus [34]. 

However, more intriguing results were presented in an exploratory analysis of the RUBY1 trial showing the efficacy of dostarlimab in p53abn tumors: HR = 0.55 (95% CI, 0.30–0.99) for PFS and HR = 0.41 (95% CI, 0.20–0.82) for OS. Of note, the NSMP subgroup did not have any increased benefit with the addition of dostarlimab to CP [26]. However, more recently, a pre-planned analysis of the MITO-END3 trial demonstrated that the efficacy of avelumab was worst in the MSS and p53-abnormal population. Despite the poor efficacy of immunotherapy in p53abn EC patients not being fully clear, the hypothesis was related to the development of hyperprogression or an immune-escape microenvironment [28].

A hypothesis that generated a randomized, double-blind, placebo-controlled phase II trial (the UTOLA study) with olaparib as the maintenance treatment after a first-line platinum-based chemotherapy failed to reach its primary endpoint (PFS in the ITT).

However, a prespecified PFS analysis according to HRD and p53 status was presented. In the HRD-positive population, defined by the number of large genomic events, median PFS was statistically longer with olaparib, regardless of p53 status [35]. This trial opened the possibility of the use of PARP inhibitors as a maintenance treatment in EC, also considering that several studies have indicated a considerable rate of BRCA mutations among patients with serous EC and an increased risk of EC with serous histology among BRCA mutation carriers [36]. 

## 7. Treatment Strategies for EC in the Near-Future

The DUO-E trial evaluated the addition of an anti PD-L1 antibody (durvalumab) to standard first-line chemotherapy, followed by a maintenance therapy with durvalumab and olaparib/placebo. In the ITT population, the arm durvalumab plus olaparib/placebo demonstrated a longer PFS than the standard arm. In the prespecified subgroup analysis by MMR status, patients treated with durvalumab alone or durvalumab plus olaparib showed a similar HR: 0.42 (95% CI, 0.22–0.80) and 0.41 (95% CI, 0.21–0.75), respectively. A slight difference in favor of the combination of immunotherapy and PARP inhibitor versus durvalumab alone was observed in the pMMR population (HR = 0.76, 95% CI = 0.59–0.99). It has to be underlined that in this trial, about 70% of the patients enrolled underwent an HRR status evaluation using the Foundation One CDx NGS assay. HHR-mutated patients were defined with the presence of deleterious or suspected mutations in ATM, BRCA1, BRCA2, BARD1, BRIP1, CDK12, CHECK1, CHECK2, FANCL, PALB2, RAD51B, RAD51C, and RAD54L. However, in the subgroup analysis of PFS for the patients treated with durvalumab and olaparib versus the control, the benefit was consistent for all the subgroups (HRRm, non-HRRm, and unknown HRR status) [37]. 

Similarly, the primary analysis of PFS of part 2 of the RUBY trial was recently presented, investigating the addition of niraparib to dostarlimab in the maintenance setting. In the overall population, dostarlimab plus chemotherapy followed by dostarlimab plus niraparib compared to placebo plus chemotherapy followed by placebo showed a statistically significant reduction in the risk of disease progression or death (HR = 0.60, 95% CI = 0.43–0.82), with a clinically meaningful improvement of 6.2 months in median PFS (14.5 months vs. 8.3 months). In the pMMR/MSS population, the results were similar with an improvement in median PFS (14.3 months vs. 8.3 months; HR = 0.63; 95% CI = 0.44–0.91) [38]. 

## 8. Other Treatment Strategies for EC

### 8.1. Target-Driven Treatments

Prior to the TCGA classification, a dualistic model for the development and progression of EC divided these tumors into type I and type II based on biologic, molecular, and clinical parameters. Type I, mainly endometrioid histology, comprises 80% of cases, is thought to arise from persistent unopposed estrogen stimulation, and is generally estrogen receptor (ER)- and progesterone receptor (PR)-positive. Genetic alterations associated with these tumors include microsatellite instability (20% to 40%), PTEN deletions or mutations (50% to 80%), PIK3CA mutations (30%) and amplification (2% to 14%), the activation of K-ras (15% to 30%), and gain-of-function mutations in b-catenin (25% to 40%) [4]. In contrast to type I EC, type II is often represented by non-endometrioid histologies, such as serous and clear cell carcinomas. A lack of an association with excess endogenous or exogenous estrogen was also seen. These tumors generally occur in postmenopausal women in the setting of an atrophic endometrium. Other potential risk factors for type II ECs include obesity and diabetes. A personal history of breast cancer and being a BRCA1 mutation carrier may also be risk factors for developing serous histology endometrial tumors. Type II tumors behave much more aggressively and show a propensity for deep infiltration, lymphovascular invasion, and distant spread. Again, ER and PR are generally negative or weakly positive [39]. The molecular genetic profile for these tumors is distinctly different to type I tumors and is associated with aneuploidy, p53 mutations (80% to 90%), an overexpression of HER-2/neu (8 to 13.2%), p16 inactivation (40%), PIK3CA mutations (20%) and amplification (46%), and E-cadherin alterations (60% to 90%). HER2 amplification is rare in endometrioid EC, especially in the absence of TP53 mutations, but is frequently seen among patients with uterine serous carcinoma, uterine clear cell carcinoma, and uterine carcinosarcoma [40]. Given this, trastuzumab in combination with CP was explored in a phase II trial in stage III and IV HER2-positive uterine serous carcinoma. The addition of trastuzumab as a primary treatment demonstrated a significantly longer median PFS (17.7 vs. 9.3 months, *p* = 0.015) and OS (29.6 vs. 24.4 months, *p* = 0.041), with no differences in toxicities between arms [41]. These results lead to the approval of CP plus trastuzumab as a first-line treatment for HER2-positive serous/p53-mutated endometrial cancers. 

Antibody–drug conjugates (ADCs), formed by a high-affinity antibody and a highly cytotoxic payload connected by a stable linker, combine the specificity of monoclonal antibodies with the anti-tumor activity of potent cytotoxic drugs. For EC, antibody–drug conjugates offer a promising therapeutic option. Trastuzumab deruxtecan, a monoclonal antibody targeting HER2, combined with a topoisomerase I inhibitor, showed an ORR of 57.5% in EC patients, regardless of HER2 expression, that reached 84.6% in patients with HER2 3+ EC tumors [42]. Similarly, in endometrial carcinosarcoma tumors, regardless of HER2 expression intensity, the disease control rate was 100%, and the median PFS was 6.7 months [43].

Another interesting ADC under investigation is sacituzumb govitecan, a humanized monoclonal antibody targeting trophoblast cell-surface antigen-2 (TROP2) linked to a topoisomerase I inhibitor. An ORR of 25% and a clinical benefit rate (CR, partial response and stabilization of disease ≥ 6 months) were observed in an early analysis of 28 patients with EC treated within the phase II TROPiCS-03 trial [44]. An increasing number of ADCs are under clinical investigation in EC, targeting HER2, folate receptor alpha (FRα), TROP2, and B7-H4, and may reach the clinic in the near-future [45]. 

### 8.2. Hormonal Therapy With or Without Targeting Agents

Given that type I EC is associated with a state of hyperestrogenism, hormonal therapy has been assessed in its management. In the normal endometrium, progesterone antagonizes the actions of estrogen and inhibits estrogen-induced cell proliferation. Progestins, such as medroxyprogesterone acetate, have been used as a fertility-sparing treatment for grade 1 EC in premenopausal women or in those women considered poor operative candidates, with a high rate of complete remission (82.8%, 95% CI, 72.3–91.2) [8]. Despite the high response rate in the primary setting, medroxyprogesterone acetate has shown modest activity for recurrent or advanced EC, with a median PFS and OS of 3.2 months and 11.1 months, respectively [46]. 

Many other hormonal therapies have been tested in advanced and recurrent EC, such as aromatase inhibitors (exemestane, letrozole, and anastrozole), fulvestrant, and gonadotropin-releasing hormone analogs. Tamoxifen alone showed a modest response rate of 10–46% [47], and it has to be considered that tamoxifen is a weak estrogen agonist in the endometrium. Aromatase inhibitors serve to block the peripheral conversion of androgens to estrogens and, although they have good tolerability, they showed low response rates (about 10%) with similar PFSs and OSs than progestins [48]. 

Given the modest activity of hormonal treatment alone, combination strategies have been evaluated. In this context, in order to leverage the crosstalk between the PI3K-mTOR pathway, the combination between everolimus and letrozole was evaluated, with an ORR of 32% [49,50]. More recently, combination therapies with aromatase inhibitors and CDK4/6 inhibitors have been studied with promising results and a median PFS ranging from about 6 months to 9 months [51,52,53]. Despite these promising results, no definitive phase III study has been conducted to date, and no CDK4/6 inhibitors are currently approved for advanced EC [54].

## 9. Discussion

Among the gynecologic malignancies, EC is currently the only gynecological cancer experiencing an increasing incidence and mortality rate. Patients diagnosed in an early stage typically have a favorable prognosis; however, those identified in a later stage have a 5-year survival rate of merely 17%, because of the few treatment options. The backbone of first-line treatment is usually based on CP and there was no standard second-line therapy following platinum failure [6,55,56,57].

The KEYNOTE-775 study confirmed the superiority of the combo pembrolizumab plus lenvatinib both in the dMMR/MSI and pMMR/MSS population. Considering only the dMMR/MSI population, pembrolizumab monotherapy in the KEYNOTE-158 study achieved a PFS of 13.1 months (95% CI, 4.3–34.4), whereas in the KEYNOTE-775 study, the PFS was 10.7 months (95% CI, 5.6-NR) in this population.

In all the three studies of immunotherapy in EC pretreated with platinum-based therapy, some dMMR/MSI EC patients did not respond to ICI alone, whereas some pMMR/MSS EC patients did. Recent progress in genomic analysis using next-generation sequencing (NGS) technology has revealed a higher average of somatic mutations in MSI cancers compared to MSS cancers. Tumor mutational burden (TMB), and the consequent mutation-associated neoantigen load, is suitable as a promising predictive biomarker of benefit for ICI therapy [58]. TMB is defined as the total number of somatic coding mutations, gene insertions, base substitutions, and deletion errors detected per million bases. Currently, the mechanism of predicting TMB tumor immune response is not fully understood. However, it is commonly accepted that a higher TMB is associated with the generation of more neoantigens, which enhances the body’s immunogenicity. This increased immunogenicity may enable tumor-specific T cells to recognize novel antigens and ultimately elicit an immune response [59,60]. In EC, it was observed that compared with the low-TMB group, activated CD4+ T cells, plasma cells, and CD8+ T cells exhibited a more abundant density in the high-TMB group [61] and the expression of regulatory T cells might mediate T cell immune suppression within the cancer milieu and thus correlate with EC progression [62]. In the KEYNOTE-158 study, the analysis of TMB in dMMR/MSI EC demonstrated that patients who had dMMR/MSI EC and high TMB had a better ORR compared to those with low TMB (ORR, 46.6% vs. 6%) [14]. In the GARNET study, regardless of the MMR status, patients who had a high TMB had a higher ORR (dMMR/MSI/h-TMB group ORR of 44.8% vs. pMMR/MSS/l-TMB group ORR of 45.5%) [16,63].

The standard second-line treatment for pMMR/MSS EC patients is the association between lenvatinib and pembrolizumab. However, the KEYNOTE-775 trial lacks the lenvatinib monotherapy comparator arm, potentially because single-agent lenvatinib demonstrated limited efficacy in second-line treatment for recurrent EC (ORR, 14.3% [95% CI, 8.8 to 21.4]) [29]. Due to the toxicity profile of the combo pembrolizumab plus lenvatinib (33% of patients with any-grade TEAEs leading to discontinuation in this arm), and the high efficacy of the immunotherapy alone in the dMMR population, dostarlimab or pembrolizumab only may be the preferred option in this setting. Some studies are ongoing to answer the question of how we can expand treatment beyond the biomarker-selected EC population. These studies analyzed the combination of immunotherapy with PARP inhibitors (PARPis). This combo seems to increase tumor-infiltrating lymphocytes (TILs) and enhance DNA damage with increased CD8+ T cells. The DUO-E trial showed a median PFS of 15.1 months (range of 12.6–20.7) in the durvalumab plus olaparib arm, a PFS of 9.6 months (range of 9.0–9.9) in the control arm, and a PFS of 10.2 months (range of 9.7–14.7) in the durvalumab arm [25]. Subgroup analysis by MMR status did not find statistical differences between durvalumab alone or durvalumab plus olaparib (HR, 0.97), but in the group of PD-L1-positive EC (TAP ≥ 1, 69% of the population), the combination durvalumab plus olaparib seems to perform better than durvalumab alone (HR, 0.67). This analysis needs to be confirmed at a longer FU. The RUBY-2 trial (a multicenter phase III study evaluating the efficacy and safety of dostarlimab plus carboplatin-paclitaxel followed by dostarlimab plus niraparib) confirmed the efficacy of the combination of dostarlimab plus niraparib [38].

In the dMMR population of EC, we are awaiting two potentially practice-changing trials in first-line EC: KEYNOTE-C93/GOG3064/ENGOT-en15 (a phase III trial of pembrolizumab alone versus platinum doublet chemotherapy in first-line dMMR/MSI advanced or recurrent EC not treated with prior chemotherapy) and ENGOT-en13/GINECO/DOMENICA (a phase III trial comparing chemotherapy alone versus dostarlimab in first-line dMMR/MSI advanced or recurrent EC). If these trials are positive, we will probably have a chemo-free future for our patients. 

The possible actual therapeutic algorithm for EC is summarized in Figure 1. Although a direct comparison between different regimens is lacking, subgroup analyses according to molecular classification may provide some suggestions as to how to choose different regimens. For instance, PFS in the exploratory analyses of the RUBY trial showed a better PFS in the dostarlimab arm for the dMMR/MSI group and for p53abn [25]. However, the secondary analyses of MITO-END3 showed different results, with avelumab ineffective in patients carrying a mutation of TP53 [28]. So, more studies in this setting are needed. Patients with POLE-mut EC seem to have a very good prognosis regardless of the treatment received, also in advanced stages. Finally, the NSMP subgroup is a very heterogeneous group of patients that need to be studied in depth. In the exploratory analyses of the RUBY trial, this group seemed to not benefit from dostarlimab (HR, 0.77; 95% CI, 0.55–1.07) [25].

So, for POLE-mut EC patients, we can consider avoiding treatments in the early stage, awaiting more data in order to consider shifting this option for advanced stages and considering the combination of chemotherapy plus ICI for now. For dMMR EC, the use of ICI reflects the earlier the better: in combination with chemotherapy or as monotherapy for patients already treated with platinum-based chemotherapy. The open question is whether it is better to use ICI alone or in combination with chemotherapy for those patients who progressed after at least 6 months to previous platinum-based chemotherapy. In both regimens (ICI alone or in combination), we show efficacy and safety in the registration trials, so to make a decision in our clinical practice, we have to take into account other parameters such as patient ECOG performance status, comorbidities, fragilities, and, last but not least, patient preference. For pMMR EC, the p53 status can guide therapeutic decisions: for p53abn EC chemotherapy plus ICI, according to RUBY subgroup analysis, awaiting more studies in this subgroup of patients can be the choice, instead of p53 wt where we can evaluate alternative target therapies (for HER 2-positive serous EC—trastuzumab; for frail patients with hormone receptor-positive EC—endocrine treatment).

p53 wild-type EC is an unmet clinical need. On the basis of SIENDO trial results presented at the ASCO meeting in June 2023 (in TP53 wild-type EC patients, selinexor delayed cancer growth for a median of 27.4 months versus 5.2 months with placebo), the X-PORT 042 trial is now ongoing—a phase III trial of selinexor including p53 wt advanced/metastatic EC. These results can change this algorithm in the future. 

The second line or latter lines, after the use of ICI, are also an unmet clinical need, and chemotherapy is the standard for fit patients. Many trials using ADC are now ongoing with promising results. 

## 10. Limitations 

The treatment landscape of EC is rapidly changing, as well as knowledge on the biology of the disease. Since the incorporation of the TGCA classification into the treatment decision process, at least for the early stages, the trials published until now have become old. In fact, all the trials on immunotherapy have tried to evaluate the efficacy of these drugs only considering EC as a dualist model (dMMR vs. pMMR), even if we are now aware of the four molecular subgroups. Thus, also in our proposed algorithm, we extrapolated data presented after post hoc subgroup analyses, with all the intrinsic limitations. We hope in the near-future to see trials that evaluate experimental drugs considering the molecular subgroups at least as stratification factors.

All the trials presented in the current review were recently published but lack long-term OS data. It has to be taken into account that EC patients frequently have other comorbidities whose treatment may be difficult to choose. In this context, with the possibility of more treatment options, in particular, the possibility to treat dMMR EC patients with immunotherapy only, avoiding the need of chemotherapy could be an important advancement. However, we are still waiting on the KEYNOTE-C93 and DOMENICA trials to confirm this opportunity.

Finally, quality of life (QoL) data were presented for the main published trials, with all the studies underlining that the new combination did not worsen the QoL. Nevertheless, a deeper analysis of the QoL data is needed, again analyzing the effect on this aspect per molecular subtype.

## 11. Conclusions

In conclusion, similarly to what took place in early-stage EC patients with the integration of a multidisciplinary approach, the TGCA classification, and technological advancements (such as multi-modal sentinel lymph node mapping) [64], the treatment paradigm for advanced EC has changed from standard CP for all to a more personalized approach. Currently, EC patients should be characterized molecularly to have prognostic and predictive information. Therapy should be prescribed while taking into account the molecular classification, disease history, and patient preferences. Evidence-based guidelines and clinical decision tools based on molecular classification are available to help physicians in treatment decision making only for the early stages. Specific recommendations for the treatment of relapsed/metastatic EC patients according to molecular classification are lacking and could be very helpful for the treating physicians. In the near-future, more treatment options will be available, further changing the therapeutic landscape of advanced EC.

## Figures and Tables

**Figure 1 ijms-25-11448-f001:**
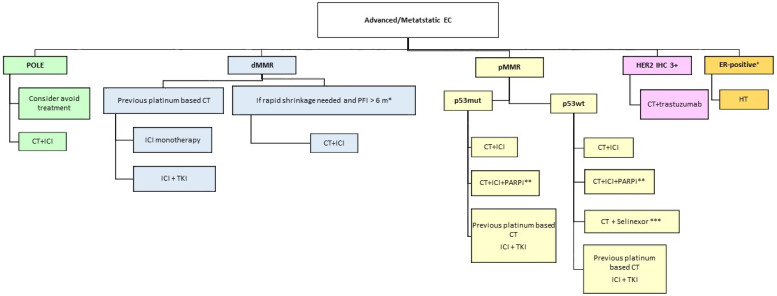
Algorithm for treating advanced/metastatic endometrial cancer.EC = endometrial cancer, pMMR = proficient mismatch repair, dMMR = deficient mismatch repair, wt = wild type, mut = mutated, CT = chemotherapy, ICI = immune checkpoint inhibitor, m = months, ER = estrogen receptor, HT = hormonal treatment. * platinum free interval borrowed from ovarian cancer and not universally recognized; ** waiting for other predictive biomarkers which may help in patients’ selection; *** waiting for the confirmatory phase III trial currently recruiting; ^+^ slowly progressive disease/frail patient.

**Table 1 ijms-25-11448-t001:** Main studies exploring immunotherapy in II line treatment for recurrent, or metastatic endometrial cancer.

Study	KEYNOTE-158	GARNET	KEYNOTE-775
Drug(s)	Pembrolizumab	Dostarlimab	Pembrolizumab plus lenvatinib
Type of study	Phase II	Phase I (part 2B)	Phase III
N. of sites	38	123	167
N. of Countries	15	NA	21
Randomization	Not randomized	Not randomized	1:1
N. of pt enrolled	90 (cohort D plus cohort K)	129 (cohort A1: dMMR) 161 (cohort A2: pMMR)	827
Primary EPs	ORR	ORR, DOR	PFS, OS
Secondary EPs	DOR, PFS, OS and safety	irORR, irDCR, irDOR, DCR, safety	ORR, safety, HRQoL, pharmacokinetics
Duration of treatment	35 cycles (2 years)	Until PD	Until PD
Hystotypes	All histological subtypes except sarcoma and mesenchymal tumors	All histological subtypes except sarcoma and carcinosarcoma	All histological subtypes except sarcoma and carcinosarcoma
FU	42.6 months	16.3 months (cohort A1) 11.6 months (cohort A2)	12.2 months (lenvatinib–pembrolizumab) 10.7 months (chemotherapy)
ORR	48% (95% CI 37–60)	43.5% (95% CI 34–53) (cohort A1: dMMR) 14.1% (95% CI 0.1–20.6) (cohort A2: pMMR)	Pembro plus lenva pMMR: 30.3% all-comers 31.9%
PFS	13.1 months (95% CI, 4.3–34.4)	immature	Pembro plus lenva - pMMR 6.6 months (95% CI 5.6–7.4) - all-comers 7.2 months (95% CI 5.7–7.6) - dMMR 10.7 months (95% CI 5.6-NR)
OS	NR (95% CI, 27.2-NR)	immature	Pembro plus lenva - pMMR 17.4 months (95% CI 14.2–19.9) - all-comers 18.3 months (95% CI 15.2–20.5) - dMMR NR

N = number, NA = not applicable, pt = patients, EPs = endpoints, ORR = overall response rate, DOR = duration of response, PFS = progression free survival, OS = overall survival, HRQoL = health-related quality of life, irORR = immune-related overall response rate, irDCR = immune-related disease control rate, irDOR = immune-related duration of response, DCR = disease control rate, PD = progression of disease, FU = follow-up, pMMR = proficient mismatch repair, dMMR = deficient mismatch repair, NR = not reached, CI = confidence interval.

**Table 2 ijms-25-11448-t002:** Main studies exploring immunotherapy in combination with chemotherapy for endometrial cancer.

Study	NRG-018	RUBY	AtTEnd	MITO-END 3
Drug(s)	Pembrolizumab	Dostarlimab	Atezolizumab	Avelumab
Type of study	Phase III	Phase III	Phase III	Phase II
N. of sites	395	113	89	31
N. of Countries	4	9	10	1
Randomization	1:1	1:1	1:2	1:1
N. of pt enrolled	816	494	550	125
Primary EPs	PFS	PFS, OS	PFS, OS	PFS
Secondary EPs	Safety, OS, QoL	Safety, ORR, disease control, response duration, time to second progressive disease, QoL, pharmacokinetic and immunogenicity analyses	Safety, ORR, disease control, response duration	Safety, ORR, OS changes in PRO scores
Duration of treatment	84 weeks (up to 14 cycles)	3 years	Until PD or unacceptable toxicity	Until PD or unacceptable toxicity
Hystotypes	Carcinosarcoma not included	Carcinosarcoma included	Carcinosarcoma included	Carcinosarcoma not included
FU	12 months dMMR/MSI 7.9 months pMMR/MSS	24.8 months dMMR/MSI 25.4 months pMMR/MSS	26.2 months dMMR/MSI 28.3 months all comers	23.3 months all-comers
Stratification factors	MMR status ECOG PS Prior cht (yes vs. not)	MMR status Prior RT (yes vs. not) Disease status	histotylogy recurrent disease vs. new diagnosis MMR status	Center Histology (serous and clear cell vs. other) Disease status
PFS	pMMR: 13.1 months dMMR: NR	pMMR: PFS at 24 months 28.4% M = dMMR: PFS at 24 months 61.4%	dMMR: NR all comers: 10.1 months	9.9 months all comers
OS	Immature data	pMMR *: OS at 24 months 66.7% dMMR *: OS at 24 months 71.3%	all comers: 38.7 months dMMR: NR	Immature data

N = number, NA = not evaluable, pt = patients, EPs = endpoints, PFS = progression free survival, OS = overall survival, PD = progression of disease, FU = follow-up, pMMR = proficient mismatch repair, dMMR = deficient mismatch repair, cht = chemotherapy, RT = radiotherapy, CI = confidence interval, QoL = quality of life, NR = not reached, PRO = patient-reported outcome, * 43% maturity of data.

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
