# Peer review of "Treatment Strategies for Advanced Endometrial Cancer According to Molecular Classification"

_ijms, 2024, doi:10.3390/ijms252111448_

Round 1

Reviewer 1 Report

Comments and Suggestions for Authors

Dear Author,

Thank you for the opportunity to review your manuscript titled "Treatment Strategies for Advanced Endometrial Cancer According to Molecular Classification." Your work addresses a critical area in the evolving management of advanced endometrial cancer (EC) and highlights the shift towards a more personalized treatment approach. While your review effectively summarizes recent advancements, there are areas that could be enhanced. Below, I outline several limitations and potential modifications to improve your manuscript. The treatment landscape of advanced endometrial cancer is changing rapidly and this review seeks to summarize the most recent evidence in this field. The review is well written and comprehensive. However, I suggest you to move the paragraph on the efficacy of pembrolizumab and lenvatinib in the dMMR population to the right section and provide a suggestion to readers on how to choose between mono-immunotherapy and immuno+TKI after a progression to a platinum-based regimen. The conclusion briefly summarizes the shift towards personalized medicine; however, it could be strengthened with specific examples of emerging therapies or technologies that may impact future treatment paradigms. References should be updated with the following citation: doi: 10.3390/healthcare12171752.

Comments on the Quality of English Language

The paper is clear and well written, a minor revision with a native-speaker could improve the quality of the paper

Author Response

Thank you for the opportunity to review your manuscript titled "Treatment Strategies for Advanced Endometrial Cancer According to Molecular Classification." Your work addresses a critical area in the evolving management of advanced endometrial cancer (EC) and highlights the shift towards a more personalized treatment approach. While your review effectively summarizes recent advancements, there are areas that could be enhanced. Below, I outline several limitations and potential modifications to improve your manuscript. 

The treatment landscape of advanced endometrial cancer is changing rapidly and this review seeks to summarize the most recent evidence in this field. The review is well written and comprehensive. However, I suggest you to move the paragraph on the efficacy of pembrolizumab and lenvatinib in the dMMR population to the right section and provide a suggestion to readers on how to choose between mono-immunotherapy and immuno+TKI after a progression to a platinum-based regimen. 

Authors reply: We changed our review accordingly and shared our opinion on which patient could benefit most from this kind of treatment.

The conclusion briefly summarizes the shift towards personalized medicine; however, it could be strengthened with specific examples of emerging therapies or technologies that may impact future treatment paradigms. References should be updated with the following citation: doi: 10.3390/healthcare12171752.

Authors reply: We have added a short statement of the progress achieved in the early stages with the integration of new knowledge and technologies.

Reviewer 2 Report

Comments and Suggestions for Authors

I have reviewed the review article titled "Treatment Strategies for Advanced Endometrial Cancer according to Molecular Classification"* presented by Tuninetti et al. The article addresses a relevant topic by discussing treatment strategies for advanced endometrial cancer (EC), integrating the latest molecular classifications. The review concludes that the treatment of advanced EC is shifting towards a more personalized approach. However, more specific recommendations could be offered regarding future research areas, such as exploring additional biomarkers or assessing the impact of treatments on patients with different genomic profiles.

Nevertheless, the context could benefit from a more extensive discussion on how this classification has improved daily clinical practice. Additionally, it would be important to further mention the limitations of previous treatments and how new therapies are driving changes in clinical outcomes.

Major Comments

-The manuscript mentions some relevant clinical trials but lacks a deep discussion on the current limitations of these trials and the need for more randomized studies that include different molecular subtypes of endometrial cancer. It would be advisable to include a section dedicated to the limitations of both the reviewed studies and the review itself.

-While the molecular classification based on TCGA is well described, I believe the analysis of the predictive utility of this classification beyond progression-free survival (PFS) and overall survival (OS) could be expanded. I recommend a more detailed discussion on the biological implications of each molecular subtype and how it influences the selection of targeted therapies and immunotherapy.

-Although the manuscript mentions various clinical trials, such as KEYNOTE-158 and RUBY1, there is a lack of direct comparisons between studies for specific molecular subtypes. It is important to add a more detailed analysis of how different regimens (such as the combination of immunotherapy and chemotherapy) perform based on molecular classification, as this could offer a more comprehensive perspective.

-The review does not address how factors such as comorbidities or prior chemotherapy responses influence patient outcomes, which limits the clinical applicability of the described strategies. It is important to highlight that several key studies have not yet provided full long-term overall survival (OS) data. I recommend explicitly mentioning this limitation and how the long-term results of these trials could impact future clinical practice. Additionally, discussing the quality of life data for patients treated with these new therapies would be valuable.

-While the manuscript provides a clear overview of the available therapeutic options, specific recommendations are lacking on how clinicians could integrate these new options into daily practice. This could be discussed by suggesting evidence-based guidelines or algorithms for therapeutic decision-making according to molecular classification, which would increase the article's utility for treating physicians.

-Expand the discussion on toxicity and serious adverse events associated with newer treatments, such as the combination of pembrolizumab and lenvatinib.

In summary, while the manuscript is well-structured and addresses a relevant topic, many full phrases are taken verbatim from the references, such as "EC remains the only gynecologic malignancy with a rising incidence and mortality. While patients diagnosed at an early stage (low-risk) have an excellent prognosis, those diagnosed at a late stage have a 5-year survival rate of only 17%" (https://nsgo.org/wp-content/uploads/2023/06/Mirza-Lecture.pdf), and similarly in other parts of the review. Furthermore, the review could benefit from deeper insights into clinical and comparative aspects. With these improvements, the manuscript would be better positioned for potential publication.

Author Response

I have reviewed the review article titled "Treatment Strategies for Advanced Endometrial Cancer according to Molecular Classification"* presented by Tuninetti et al. The article addresses a relevant topic by discussing treatment strategies for advanced endometrial cancer (EC), integrating the latest molecular classifications. The review concludes that the treatment of advanced EC is shifting towards a more personalized approach. However, more specific recommendations could be offered regarding future research areas, such as exploring additional biomarkers or assessing the impact of treatments on patients with different genomic profiles.

Nevertheless, the context could benefit from a more extensive discussion on how this classification has improved daily clinical practice. Additionally, it would be important to further mention the limitations of previous treatments and how new therapies are driving changes in clinical outcomes.

Major Comments

-The manuscript mentions some relevant clinical trials but lacks a deep discussion on the current limitations of these trials and the need for more randomized studies that include different molecular subtypes of endometrial cancer. It would be advisable to include a section dedicated to the limitations of both the reviewed studies and the review itself.

Authors reply: We perfectly agree with this suggestion, thus with added a “limitations” section in order to discuss both the limitations of the trials presented and of the review.

-While the molecular classification based on TCGA is well described, I believe the analysis of the predictive utility of this classification beyond progression-free survival (PFS) and overall survival (OS) could be expanded. I recommend a more detailed discussion on the biological implications of each molecular subtype and how it influences the selection of targeted therapies and immunotherapy.

Authors reply: We added for each section on the molecular classification a biological rationale for the treatment activity.

-Although the manuscript mentions various clinical trials, such as KEYNOTE-158 and RUBY1, there is a lack of direct comparisons between studies for specific molecular subtypes. It is important to add a more detailed analysis of how different regimens (such as the combination of immunotherapy and chemotherapy) perform based on molecular classification, as this could offer a more comprehensive perspective.

Authors reply: we have discussed this relevant topic in more detail throughout the sections of the paper and in the discussion.

-The review does not address how factors such as comorbidities or prior chemotherapy responses influence patient outcomes, which limits the clinical applicability of the described strategies. It is important to highlight that several key studies have not yet provided full long-term overall survival (OS) data. I recommend explicitly mentioning this limitation and how the long-term results of these trials could impact future clinical practice. Additionally, discussing the quality of life data for patients treated with these new therapies would be valuable.

Authors response: We included the discussion on this topic in the “limitations” section.

-While the manuscript provides a clear overview of the available therapeutic options, specific recommendations are lacking on how clinicians could integrate these new options into daily practice. This could be discussed by suggesting evidence-based guidelines or algorithms for therapeutic decision-making according to molecular classification, which would increase the article's utility for treating physicians.

Authors reply: Perfectly correct. We hope in the near future to have molecular based recommendations also for the advanced stages (suggested in the conclusions). 

-Expand the discussion on toxicity and serious adverse events associated with newer treatments, such as the combination of pembrolizumab and lenvatinib.

Authors reply: we added a description of the principal adverse events with the combination of Pembrolizumab and lenvatinib and discussed how this may influence the treatment decision process.

In summary, while the manuscript is well-structured and addresses a relevant topic, many full phrases are taken verbatim from the references, such as "EC remains the only gynecologic malignancy with a rising incidence and mortality. While patients diagnosed at an early stage (low-risk) have an excellent prognosis, those diagnosed at a late stage have a 5-year survival rate of only 17%" (https://nsgo.org/wp-content/uploads/2023/06/Mirza-Lecture.pdf), and similarly in other parts of the review. Furthermore, the review could benefit from deeper insights into clinical and comparative aspects. With these improvements, the manuscript would be better positioned for potential publication.

 Authors reply: we extensively reviewed all our manuscripts in order to improve its readability and to provide more insights in this rapidly changing landscape.

Round 2

Reviewer 2 Report

Comments and Suggestions for Authors

I believe that the authors have made significant changes to their review, adding very important information; however, this has made the reading more dense. I suggest including a final summary figure with the treatment strategies analyzed.

Author Response

Reviewer suggestion: I believe that the authors have made significant changes to their review, adding very important information; however, this has made the reading more dense. I suggest including a final summary figure with the treatment strategies analyzed.

Authors reply: We have included a figure as suggested to aid readers in the treatment decision-making process.